# Metallurgical Aspects in the Welding of Clad Pipelines—A Global Outlook

**Ivan Bunaziv \*, Vigdis Olden and Odd M. Akselsen**

SINTEF Industry, P.O. Box 4760 Torgarden, NO-7465 Trondheim, Norway

\* Correspondence: ivan.bunaziv@sintef.no; Tel.: +47-457-95-269

**Abstract:** In the present work, the metallurgical changes in the welding of clad pipelines are studied. Clad pipes consist of a complex multi-material system, with (i) the clad being stainless steel or a nickel-based superalloy, (ii) the pipe being API X60 or X65 high-strength carbon steel, and (iii) the welding wire being a nickel-based superalloy or stainless steel in the root and hot pass, with a nickel or iron buffer layer, followed by filling with carbon steel wire. Alternatively, the corrosion resistant alloy may be used only. During production of the clad pipe, at the diffusion bonding temperature, substantial material changes may occur. These are carbon diffusion from the carbon steel to the clad, followed by the formation of hard martensite at the interface on cooling. The solidification behavior and microstructure evolution in the weld metal and in the heat-affected zone are further discussed for the different material combinations. Solidification behavior was also numerically estimated to show solidification parameters and resulting solidification modes.

**Keywords:** arc welding; clad steel; nickel-based superalloys; solidification; microstructure

## 1. Introduction

Clad or lined pipes represent cheap alternative to pipes fully made of corrosion-resistant alloys (CRAs), which are very expensive materials including stainless steel or nickel-based alloys. As a substrate, a carbon steel is used. Clad pipes are widely used in refineries and petrochemical industries, fertilizer and chemical industry, power generation, environmental technology, seawater desalination, pulp and paper manufacturing, the food industry, and shipbuilding [1]. Concerning the oil and gas industry, clad and lined pipes, including the pipe-in-pipe method, have already been installed in the North Sea and the Norwegian Sea. These are often made of carbon steel with an inner clad or liner to protect the carbon steel against hydrogen embrittlement and sulfide stress corrosion cracking. The clad and liner are most frequently made of austenitic stainless steel, often 316L, or a Ni-based superalloy, typically Inconel 625. Clad pipes are mainly produced by hot roll bonding, explosion bonding, and weld overlay. Hot roll bonding is by far the most widely adopted production method [2]. Nickel-based superalloys also frequently used in the aerospace industry [3].

There are different methods of producing clad pipe, e.g., via a mechanical inner liner. This technology is a cost-effective alternative to more expensive options such as solid CRAs or metallurgically bonded clad pipe. The principles are the same as for metallurgically bonded clad pipes, but involve being connected to the external carbon steel pipe through a mechanical bond. Recent results [4] claimed that the liner separation is of minor importance and the use of the established acceptance criterion derived for plain carbon steel pipes can be justified to apply to CRA-lined pipelines. Both clad pipe and lined pipes can be laid by reeling. Although lined pipes are a cheaper solution than clad pipes, the welding technology for the former case may be more challenging. Since the welding solutions for reelable CRA-lined pipe may be considered as key enabling technologies for future exploitation of deep-water oil and gas reserves, it will not be further addressed in the present work.

Welding represents a very complicated joining process by the melting and subsequent solidification of metal forming a hermetic and strong joint. During the process, all four states of matter are present and interact. The most widely used methods are gas tungsten arc welding (GTAW) and gas metal arc welding (GMAW). In the case of GTAW, filler wire is added externally to the weld pool where the tungsten electrode is nonconsumable, while in GMAW, filler wire is added directly since it is a consumable electrode. Both are very well suited to automated processes [5–7].

The welding of clad pipelines may encounter several challenges, specifically when damage or leakage occur and the repair of clad pipes is required. Repair of a pipeline implies replacement of the damaged part with a new pipe section to be joined at the seabed via remote control. Such subsea operations are very expensive, which makes the robustness of the joining operation of crucial importance. This led to the development of the welded sleeve concept for pipe repair, which was the only technique where acceptable robustness could be achieved [8]. In a repair situation, the welded sleeve technique involves the installation of an oversized pipe segment (sleeve) over the joint by threading the pipe ends through the sleeve and performing a subsea fillet weld between the pipe and sleeve on each end [9]. This technique is fully remotely controlled and has been proven by simulating conditions in a hyperbaric chamber to work down to at least 2500 m depth, corresponding to 250 bar pressure conditions, for the welding process. However, the sleeve solution is not acceptable for clad pipes due to the direct exposure to the environment, as it may suffer from corrosion and/or hydrogen embrittlement during service [10]. A comprehensive review is available for further studies [11] with cohesive zone numerical modeling [12]. To retain the corrosion resistance of the clad pipe, the root and hot pass should be deposited with a wire compatible with the clad material. This limits the freedom in the selection of welding consumables.

In recognition of this situation, the development of hyperbaric butt-welding technology was initiated with the aim to develop a fully remotely controlled dry welding technology for deep waters (below about 200 m depth). In addition to the welding technology challenges, there are metallurgical aspects that need to be considered. Therefore, the present paper addresses the metallurgical aspects in the welding of clad pipes, involving clad materials such as AISI 316L SS and Inconel 625 (typically of 3 mm thickness) and the typical pipeline API X60/X65 high-strength carbon steels. With the addition of the specific welding wire—Ni-based superalloys such as Inconel 625 and Alloy 59—a complex multi-material system is formed that needs more comprehensive understanding. Inconel 625 filler wire has been proven (high toughness at −30 °C) for hot-tap welding operating in subsea conditions [13] in joining X65 high-strength steel pipelines. Examples from an ongoing research project will be used to illustrate various microstructure observations, concentrating on the clad, the heat-affected zone (HAZ) of the pipe and the weld metal, or the fusion zone. Possibly, for onshore operations, a relatively new method of friction drilling and form tapping can be used to reduce the harmful effect of heat on welded joints [14].

Under hyperbaric conditions for pipe repair, from a practical standpoint, the use of multiple wires will be very difficult, such as is frequently employed in conventional clad pipeline welding; i.e., a CRA alloy for the root and hot pass, followed by a buffer layer of either Fe or Ni, and finally, filler passes by a matching ferritic wire. In a fully remotely controlled hyperbaric operation, a much simpler solution with one wire only will be employed throughout the weld. The use of backing gas is applicable since it is purged into the chamber prior to welding to remove the seawater. The moisture and oxygen content of the chamber gas must be carefully controlled continuously during welding. Moreover, the use of one welding process for the root pass and another for the fillers will not be suitable. Although GTAW is frequently used in the root pass welding of clad pipes due to the smooth surfaces, it is not applicable in remotely controlled conditions. GMAW is therefore under development for future dry subsea pipe welding, including the use of an inner clad.

The manuscript is organized as follows: description of the clad pipes and their production is followed by an explanation of microstructure evolution principles (applicable for both clad pipe production and welding), applied to the weld metal and HAZ microstructure evolution with different

material combinations, a description of hyperbaric (repair in subsea level) welding conditions, and an outline of the potential numerical simulation of the arc physics.

## 2. Production of Clad Pipes

The metallurgical behavior of clad pipes depends on their production route during fabrication. Diffusion bonding represents a frequently used technique. During production, a certain temperature range is required to achieve metallurgical bonding, usually from $0.5T_m$ to $0.8T_m$ (where $T_m$ is the melting temperature). The temperature–time cycle will thus allow the diffusion of alloying elements that may influence the structural integrity of the pipe. Thus, the interface between the two materials will attain a gradient of chemical composition and hence produce other microstructures compared to the two surrounding materials, as indicated in Figure 1. One example is shown in Figure 2, which shows carbon diffusion into the 316L stainless steel (SS) clad due to the higher solubility in austenite. The carbon content at the bimetallic or clad/pipe interface on the clad side is between 0.15 and 0.20 wt %. A consequence is that extensive precipitation of $Cr_{23}C_6$ takes place [15]. It is also expected that other elements such as Ni, Cr, and Mo may diffuse to the interface, although to a lower extent than carbon due to the lower diffusivity of these large-atom elements. Due to diffusion of alloying elements, microsegregation may facilitate martensite formation at the interface. Local microhardness increases of up to 350 HV has been found, as shown in Figure 3, as an indication of martensite presence. Another influencing factor may be a deformation hardening during the hot-roll bonding process [16].

Measurements made on welded samples gave evidence that martensite was formed. The maximum microhardness values of 362 and 392 HV in the clad close to the interface for two different macros was reported [17]. Because of the carbon depletion of the pipeline steel measured by an electron probe microanalyzer (EPMA), the hardness level on the steel side close to the interface is low (156 HV). As an example, Line 3 in Figure 2 shows a carbon reduction from 0.16 to 0.03 wt % C. Moreover, some grain growth of the carbon steel close to the interface was observed, which is also consistent with the lower hardness in this region. Therefore, remarkable gradients of hardness and microstructure occur across the interface, which may potentially influence the integrity of the pipe.

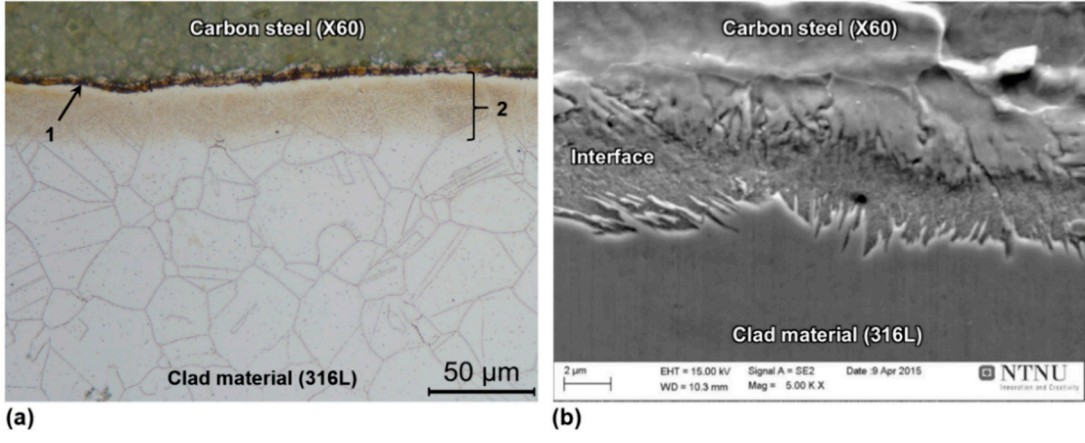

**Figure 1.** Carbon steel–clad (316L) interface microstructure as observed by (**a**) optical microscopy and (**b**) SEM; 1—the interface layer and 2—the width of the carbon diffusion layer [18].

In contrast to direct cladding with AISI 316, the application of a 30–35 µm nickel interlayer between the clad and the carbon steel reduces the carbon diffusion into the clad substantially [19].

With the Ni-based alloy 625 as the clad material, there is a higher potential for elemental segregation than with stainless steel. However, diffusion of Fe into the clad is regarded as a more critical issue. This is also the case in the cladding of carbon steel with Alloy 625. This alloy is frequently used in the overlay welding of carbon steels to improve the corrosion resistance, where the dilution with the substrate is kept low to preserve the initial corrosion resistance of the clad [20].

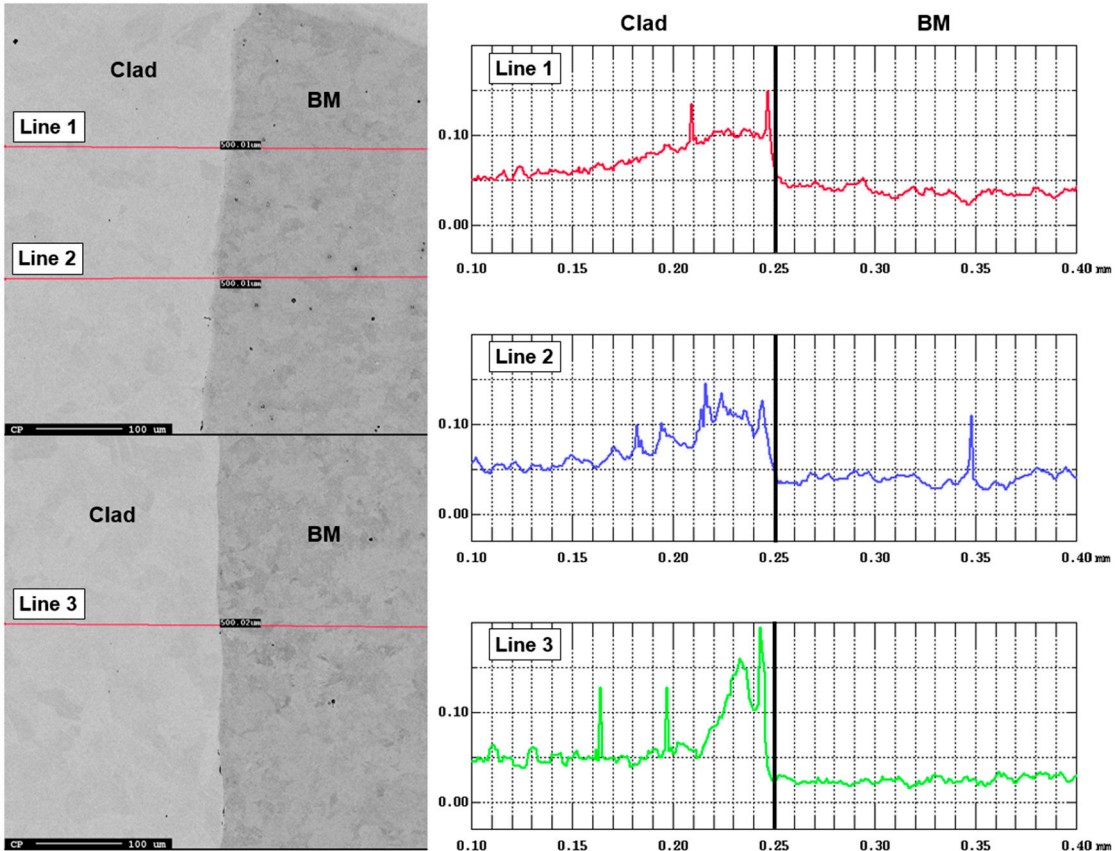

**Figure 2.** Variation of carbon content (wt %) across the clad/pipe interface [21]. Lines 1, 2, and 3 represent measurement distance and BM is base metal.

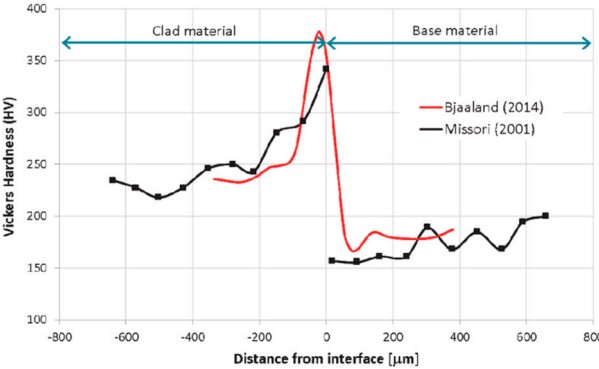

**Figure 3.** Increase in hardness values at the bimetallic interface [21].

It should be noted that the chemical composition of the weld clad/overlay must comply with the specification of the material. In the case of Alloy 625, the maximum allowable Fe content is 5% (according to API 5LD [22] specifications) or 10% (according to the DNV GL [23] standard). To achieve such low dilution, particularly the 5% requirement, the cold metal transfer (CMT) variant of the gas metal arc welding (CMT-GMAW) is an advantageous process due to lower heat input.

## 3. Microstructure Evolution

In the welding of clad pipes, it is common to use corrosion-resistant alloys as consumables (filler wire), often Alloy 59 or Inconel 625, and 304L/309/316L may be used for the root and the hot pass. Thus, the weld metal (WM) will be different from the pipe material, which is typically X60 or X65 carbon steel. Therefore, the welded joint of clad pipes consists of three or four different materials.

Hence, the microstructure and how it influences the final properties of the weldment is complex. In the following section, microstructures of the WM, HAZ, and clad will be discussed, starting with the solidification of WM.

### 3.1. Solidification

#### 3.1.1. Fundamentals of Solidification Theory

Welding of clad pipes is most often performed with an Inconel 625 or an Alloy 59 filler wire. Sometimes, AISI 309 wire is employed for root and hot pass deposition. Stainless steel solidifies in different modes according to its chemical composition. In addition, the cooling rate may induce a shift in solidification [24]. The solidification mode of stainless steel can be divided into four types according to the ratio between ferrite-($\delta$) and austenite-($\gamma$) stabilizing elements, i.e., $Cr_{eq}/Ni_{eq}$ [25]:

$$\text{F (ferritic) mode: } L \text{ (liquid phase)} \rightarrow L + \delta \rightarrow \delta \rightarrow \delta + \gamma \quad \text{for } Cr_{eq}/Ni_{eq} > 2.0 \quad \text{(1a)}$$

$$\text{FA (ferritic-austenitic) mode: } L \rightarrow L + \delta \rightarrow L + \delta + \gamma \rightarrow \delta + \gamma \rightarrow \gamma \quad \text{for } 1.5 < Cr_{eq}/Ni_{eq} < 2.0 \quad \text{(1b)}$$

$$\text{AF (austenitic-ferritic) mode: } L \rightarrow L + \gamma \rightarrow L + \delta + \gamma \rightarrow \gamma + \delta \rightarrow \gamma \quad \text{for } 1.37 < Cr_{eq}/Ni_{eq} < 1.5 \quad \text{(1c)}$$

$$\text{A (austenitic) mode: } L \rightarrow L + \gamma \rightarrow \gamma \quad \text{for } Cr_{eq}/Ni_{eq} < 1.37 \quad \text{(1d)}$$

The values of $Cr_{eq}$ and $Ni_{eq}$ can be calculated by a variety of equations. Olson [26] presented more than 15 different equations for different purposes (welding, casting) and including/excluding some of the microalloying elements, such as Cu and Co for $Ni_{eq}$ and Nb, Mo, Ti, Al, V, W, and Ta for $Cr_{eq}$. However, the authors [27] had previously used equations by Hammar and Svensson [28] to determine the solidification of stainless steel weld metals. The Cr and Ni equivalents can be calculated from Equation (2a) and Equation (2b), respectively. Values are represented by the wt % of the elements. Based on these equations, the solidified weld metal can be visualized (see Figure 4).

$$Cr_{eq} = Cr + 1.37\ Mo + 1.5\ Si + 2\ Nb + 3\ Ti \quad \text{(2a)}$$

$$Ni_{eq} = Ni + 22\ C + 14.2\ N + 0.31\ Mn + Cu \quad \text{(2b)}$$

Thus, it is indicated by extrapolation in Figure 4 that the classical austenitic stainless steels like AISI 309 SS and AISI 316 SS primarily solidify as ferrite.

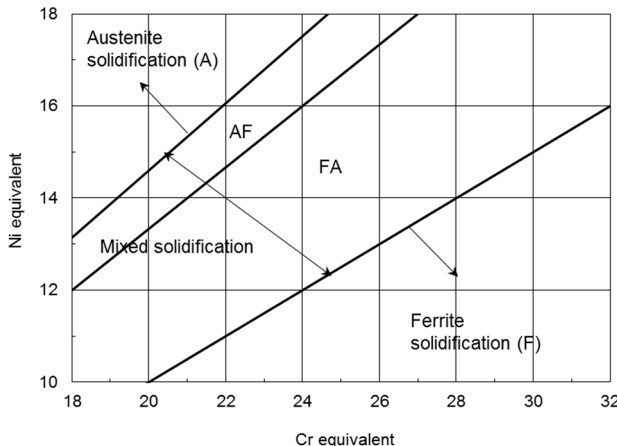

**Figure 4.** Possible solidification modes and nomenclature according to Equations (1a)–(1d).

For Alloy 59, very limited information is available in the literature, while there are numerous publications on other Ni-based alloys. However, those are mainly precipitation-hardened alloys in the 7xx series of Inconel alloys, frequently used in aerospace applications. Another frequently used

Ni-based alloy is Inconel 625, and the nominal chemical compositions of Alloy 59 and Inconel 625 are outlined in Table 1. The main difference between them is the Nb content of Inconel 625 and the much higher Mo addition in Alloy 59. The mechanical properties are also comparable, with yield strengths of ≥340 and 350 MPa for Alloy 59 and Inconel 625, respectively. Their corresponding tensile strengths are ≥690 and 770 MPa.

**Table 1.** Comparison of nominal chemical composition (wt %) of different materials.

| Alloy | Ni | Cr | Mo | Fe | Nb | Co | Mn | Ti | Si | C |
|-------|-----|-----------|-----------|-----------|-----------|------|------|------|------|-------|
| Alloy 59 | 59.0 | 22.0–24.0 | 15.0–16.0 | ≤1.5 | - | ≤0.3 | ≤0.5 | - | ≤0.1 | ≤0.01 |
| Inconel 625 | 58.0 | 20.0–23.0 | 8.0–10.0 | 5.0 | 3.15–4.15 | 1.0 | ≤0.5 | 0.4 | 0.5 | 0.1 |
| X65 | 0.5 | 0.02 | - | 98.5 | 0.05 | - | ≤1.4 | 0.04 | 0.25 | 0.07 |
| 316L | 10.0–14.0 | 16.0–18.0 | 3.0 | 63.0–64.0 | - | - | ≤1.3 | - | 0.75 | ≤0.03 |

Several experimental studies have been conducted to investigate the behavior of the solidification in superalloys containing niobium (Nb). Nb-bearing superalloys (like the popular Inconel 625/718) terminate the solidification by a eutectic-type reaction between $\gamma$- and various Nb-rich phases, such as NbC and Laves phases [29]. The creation of a crystal from an alloy melt causes a local change in the composition [30]. When the dendrites solidify, elements such as Mo and Nb, as well as impurities, segregate to interdendritic regions [31]. It was shown [32] that the presence of iron in Ni-based superalloys lowers niobium solubility in the austenite phase. When iron is in a solid solution in the $\gamma$ Ni–Cr phase, the ability of Nb to remain in solution is limited [33] and partitioning of Nb to interdendritic regions in the WM is increased. The solidification pattern follows Equation (3), where primary dendrites of $\gamma$ solidify first with a final microstructure of $\gamma$ with NbC and Laves phases precipitated in the interdendritic regions.

$$L \rightarrow L + \gamma \rightarrow L + \gamma + NbC \rightarrow L + \gamma + NbC + \text{Laves} \rightarrow \gamma + NbC + \text{Laves} \tag{3}$$

Niobium-free alloys, such as Alloy 59, solidify by a simple $L \rightarrow \gamma$ transformation without any eutectic-like reaction and exhibit a relatively narrow solidification temperature range. Solidification cracking is therefore not expected.

When compared with carbon and low-alloy steels, Ni-based alloys have a high temperature gradient (in WM) due to lower thermal conductivity, which is related to the high alloying level. Therefore, a planar solidification front may not be stable, and the primary solidification mode is cellular-dendritic, as shown in Figure 5. Here, only the grains with a crystallographic preferential growth orientation (along the (100) direction, Miller index, for metals having a cubic crystal lattice structure [34]) relationship aligned with the heat flow direction will continue to grow. When the preferential growth direction of the solid starts to deviate significantly from the heat flow direction, the growth stops, and new grains nucleate successively with the preferential growth direction nearly along the heat flow direction. The criterion for constitutional supercooling for plane front instability can be mathematically estimated [35] as follows:

The plane front will be stable when:

$$\frac{G}{R} \geq \frac{\Delta T_0}{D_L} \tag{4a}$$

Planar instability will occur when:

$$\frac{G}{R} < \frac{\Delta T_0}{D_L} \tag{4b}$$

Here, $G$ is the temperature gradient (°C/m) in the liquid, $R$ denotes the solidification front (or crystal) growth rate (μm/s), $\Delta T_0$ is the equilibrium solidification temperature range (at composition $C_0$), and $D_L$ represents the solute diffusion coefficient (m$^2$/s) in liquid. The frequently used $G/R$ ratio is a solidification morphology parameter.

In the diagram shown in Figure 5, the range of solidification modes are assumed. Carbon steel (e.g., X65) has planar solidification in a wide range of temperature gradients due to low alloy content.

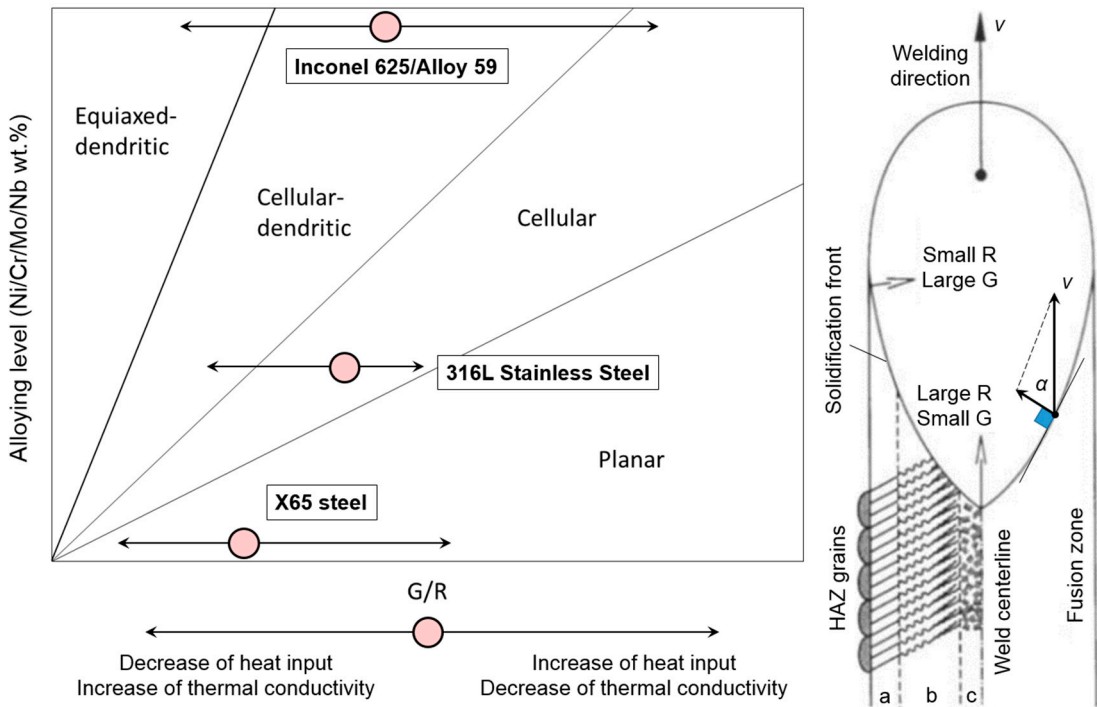

**Figure 5.** Schematic representation (modified from [34]) of the combined effect of crystal growth rate and melt temperature gradient on the weld metal solidification microstructure for different alloys; a—cellular, b—cellular-dendritic, and c—equiaxed-dendritic solidification mode.

The product *GR* is related to the cooling rate $\varepsilon$ (°C/s) as follows [36]:

$$\varepsilon = GR \tag{5}$$

This means that the dendrite spacing increases with increasing cooling time, i.e., with increasing heat input in welding.

The solidification rate parameter (*R*) can be estimated empirically based on the angle ($\alpha$) between the welding direction and solidification growth direction, or molten surface normal, as shown in Figure 5. The solidification rate (2D, the top surface) is calculated by the following equation:

$$R = v \cos \alpha \tag{6}$$

where *v* is welding speed.

The solidification rate for the 3D case can be calculated according to [37]:

$$R = v \frac{-\frac{\partial T}{\partial x}}{\sqrt{\left(\frac{\partial T}{\partial x}\right)^2 + \left(\frac{\partial T}{\partial y}\right)^2 + \left(\frac{\partial T}{\partial z}\right)^2}} \tag{7}$$

It should be noted that Equation (5) does not represent a linear relationship due to the complex heat flow in welding. However, it is assumed to be linear for simplicity to identify solidification parameters ($\varepsilon$, *G*, and *R*).

### 3.1.2. Numerical Simulation of Solidification Parameters

In order to estimate solidification parameters by finite element analysis (ABAQUS 6.14 with DFLUX subroutine programmed in Fortran), combined double Goldak's ellipsoid and surface Gaussian heat source were used [38]. Different heat sources are needed to simulate the typical *finger* penetration geometry of the GMAW welds, according to Yang and Debroy [39], since there is an impingement of droplets from the electrode into the weld pool which transports the additional heat. A detailed modeling procedure and boundary conditions can be found in [40]. Applied thermophysical parameters for the materials are outlined in Table 2. The results of numerical simulation, achieved by a *trial-and-error* method, are presented in Figure 6. The process parameters were set to 7 mm/s as the welding speed and 5.5 kW (220 A; 25 V) as the arc power. A good agreement with previous experimental results (for X70 carbon steel) welded under hyperbaric conditions of 10 bar [41] was found.

**Table 2.** Summary of thermophysical properties for Ni-based alloys and various steels.

| Material | Density (kg m$^{-3}$) | Thermal Conductivity (W m$^{-1}$ °C$^{-1}$) | Specific Heat (J kg$^{-1}$ °C$^{-1}$) | Solidus Point (°C) | Latent Heat of Fusion (J kg$^{-1}$ °C$^{-1}$) |
|---|---|---|---|---|---|
| Inconel 625/Alloy 59 [42] | 8440 | 10 | 410 | 1290 | 227000 |
| 316L SS [43] | 7200 | 19 | 712 | 1424 | 274000 |
| X65 carbon steel [44] | 7800 | 32 | 726 | 1494 | 277000 |

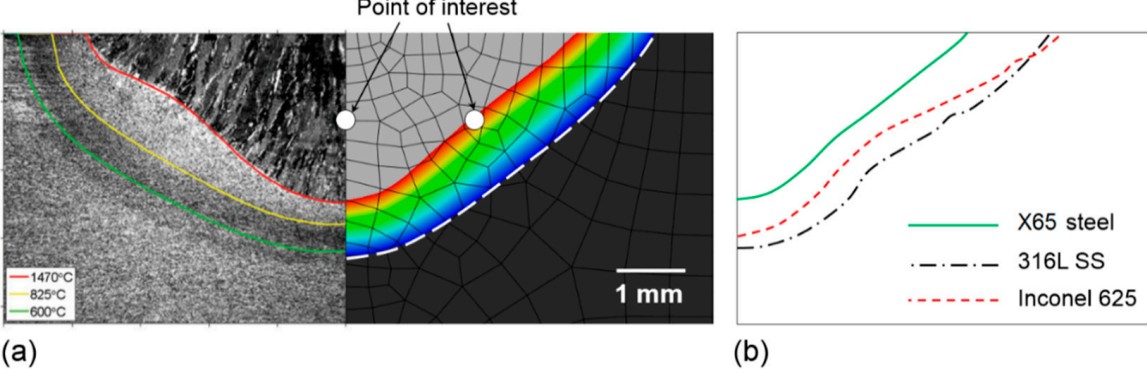

**Figure 6.** (**a**) Simulated weld of X65 carbon steel, and (**b**) fusion line geometries for different materials.

Due to different spatial temperature gradients and complex weld pool physics, the cooling rate as well as solidification rate varies according to the specific location in the WM or the HAZ [43]. Therefore, in the same WM, different solidification modes are common, as illustrated in Figure 5. At the WM centerline, the temperature gradient is smaller and the solidification rate faster when compared to the area near the FL (fusion line or liquid/solid interface) for each selected material. The most widely used parameter to describe cooling in welding for carbon steel is the cooling time from 800 °C to 500 °C (denoted as $\Delta t_{8/5}$), since most phase transformations take place within this range [45]. However, for the crystal growth rate in WM for any kind of alloy, the cooling rate from the peak temperature to the solidus line is used. The finite element analysis (FEA) results at the centerline are compiled in Table 3. The centerline is illustrated in Figure 6. The solidification parameters are outlined in Table 4 for the liquid/solid interface located at 1/4 of the weld pool length at the solidification front, and in the case of the top plane, see Figure 7. For the mid-plane case, the angle of the *G* vector is more complex to find, since it is normal to the tangential plane for a curved solid/liquid interface. Moreover, this surface is constantly in motion because it follows the moving heat source.

**Table 3.** Estimated solidification parameters at the WM centerline on top-plane. Cooling rate is calculated for temperature range from peak temperature to solidus point.

| Material | Cooling Rate, $\varepsilon$ (°C/s) | Temperature Gradient, $G$ (°C/mm) | Solidification Growth Rate, $R$ (mm/s) | F-Factor = $G/R$ |
|---|---|---|---|---|
| Inconel 625/Alloy 59 | 1122 | 160 | 7.0 | 23 |
| 316L SS | 925 | 132 | 7.0 | 19 |
| X65 carbon steel | 874 | 125 | 7.0 | 18 |

**Table 4.** Estimated solidification parameters at the liquid/solid interface, i.e., solidification front. Cooling rate is calculated for the temperature range from the peak temperature to solidus point.

| Material | Cooling Rate, $\varepsilon$ (°C/s) | Temperature Gradient, $G$ (°C/mm) | $\alpha$ Angle (Degree) | Solidification Growth Rate, $R$ (mm/s) | F-Factor = $G/R$ |
|---|---|---|---|---|---|
| Inconel 625/Alloy 59 | 323 | 242 | 79 | 1.336 | 181 |
| 316L SS | 241 | 166 | 78 | 1.455 | 114 |
| X65 carbon steel | 373 | 149 | 69 | 2.509 | 59 |

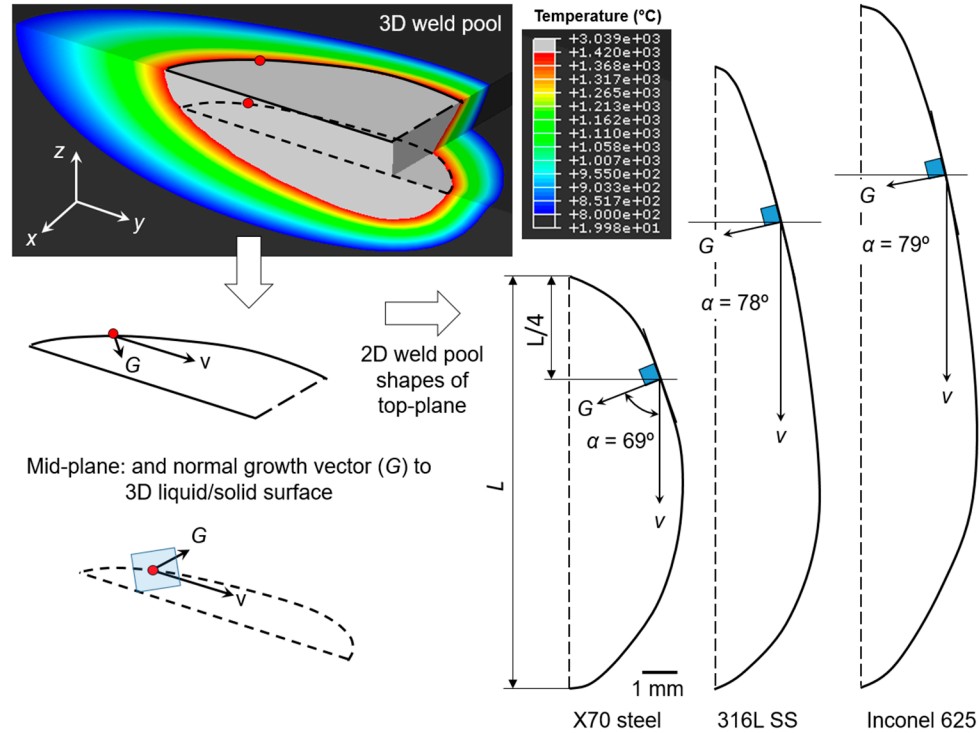

**Figure 7.** Numerically simulated weld pool profiles from a top-down view (middle plane) and identification of the solidification angle $\alpha$ at specific points.

Blecher et al. [37] established a solidification map of a Ni-based alloy (Inconel 690). For the top of the weld pool, the $R$ and $G$ values are similar to the estimated values for Inconel 625/Alloy 59 (Table 3; Table 4). Since fiber laser beam welding in the keyhole mode was applied, higher $G$ values for the deeper part of the weld pool were obtained by Wei et al. [46]. Along the weld centerline, the $R$ parameter is very similar to the welding speed, meaning that the solidification angle ($\alpha$), used in Equation (6), is approximately equal to 0 (cos $\alpha$ = 1) at the centerline, and hence $R \rightarrow v$. At the centerline, the X65 steel has a smaller $G$ value due to higher heat conduction. For the same reason, the solidification rate for X65 is the highest for the materials studied here. It is important to note that the $R$ parameter is very sensitive to the $\alpha$ angle (based on Equation (6) and Table 4).

The primary dendrite arm spacing ($\lambda_1$) is shown in Figure 8, and has a strong effect on mechanical properties [36]. Based on Kurz and Fisher [47], a generalized (simplified) formula for the primary dendrite spacing is:

$$\lambda_1 = AG^{-m}R^{-n} \tag{8}$$

here, $m$ (=0.5) and $n$ (=0.25) are constants, and $A$ is a composition-dependent (alloy) material parameter.

Since $GR$ values are obtained and $\lambda_1$ can be measured from micrographs, the $A$ parameter can be estimated based on Equation (8).

The secondary dendrite arm spacing ($\lambda_2$, Figure 8) may also strongly influence the mechanical properties. Its size will also depend on the cooling rate [36]:

$$\lambda_2 = B(GR)^{-n} \tag{9}$$

where $B$ is a material-dependent parameter and $n$ (=1/3) is a constant.

Tertiary dendrites ($\lambda_3$, Figure 8) have minor effects, and some related studies can be found in [48]. Therefore, these are not discussed in detail here due to their lower importance.

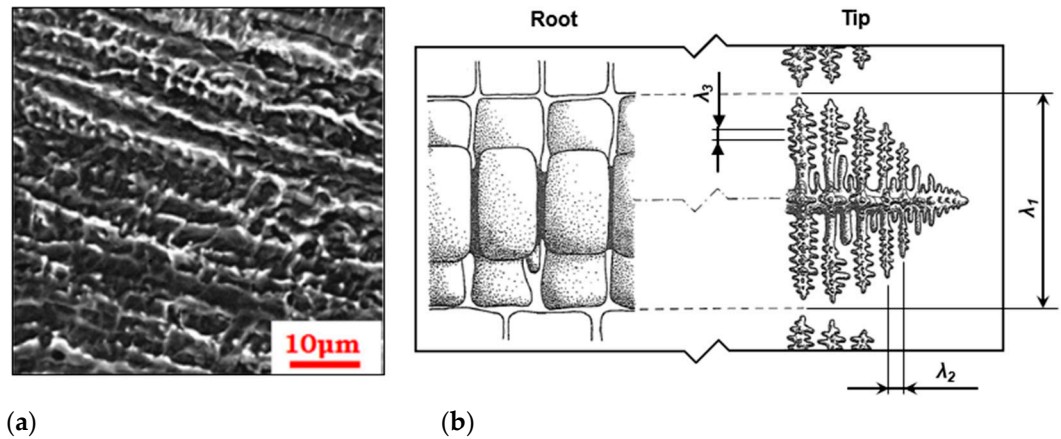

**(a)**          **(b)**

**Figure 8.** (**a**) SEM of cellular (columnar) dendritic structure in the WM of 304 SS [43] and (**b**) classification of dendritic arc spacing (modified from [49]). $\lambda_1$, $\lambda_2$, and $\lambda_3$ indicate primary, secondary, and tertiary dendrite arm spacing, respectively.

### 3.2. Segregation in Welds

Segregation is a defect in welded joints, where the alloying elements are enriched within specific areas due to motion of liquid and can decrease mechanical properties locally. Segregation is very common in stainless steels and Ni-based superalloys due to high alloy content, whereas low segregation tendency in low-carbon steel is observed, except for the segregation of impurities (S, P).

The Scheil equations, or non-equilibrium level rule, describe the solute redistribution during solidification of an alloy [50]:

$$C_S = kC_0(1 - f_S)^{(k-1)} \tag{10}$$

and

$$C_L = C_0 f_L^{(k-1)} \tag{11}$$

where $C_S$ is the element concentration in the solid, $C_L$ is the concentration in liquid, $C_0$ represents the initial concentration in the liquid, $k$ is the effective segregation coefficient ($\leq 1.0$), and $f_S$ and $f_L$ denote the volume fraction of solid and liquid, respectively.

The segregation coefficient can be estimated as follows [50]:

$$k = \frac{C_S}{C_L} \tag{12}$$

The segregation coefficient *k* can also be applied in determination of microsegregation in solidification, i.e., the distribution of elements between the core of primary dendrites and the interdendritic areas. Based on Figure 9, it is shown that Mo and Nb segregate to the interdendritic region, while Ni and to some extent Cr are enriched in the dendrites. Silva et al. [51] reported that the constant *k* for Ni and Fe had values just above 1.0, which indicates a slight segregation into the dendrite core. A coefficient of 1.0 implies that no segregation occurs. In the case of Mo, the coefficient is ~0.9, resulting in segregation to the liquid metal and hence enriching the interdendritic region. Like Mo, Nb also segregates to the liquid, but with a greater intensity due to a lower *k* value of ~0.5. Therefore, this strong segregation of Nb has been appointed as reason for the formation of the secondary phases such as the Laves phase in welding with Alloy 625 wire. For the use of Alloy 59 wire, this will not be the case due to the absence of Nb. The extensive segregation in high-alloy materials can cause residual stresses during solidification and therefore reduced mechanical properties [52].

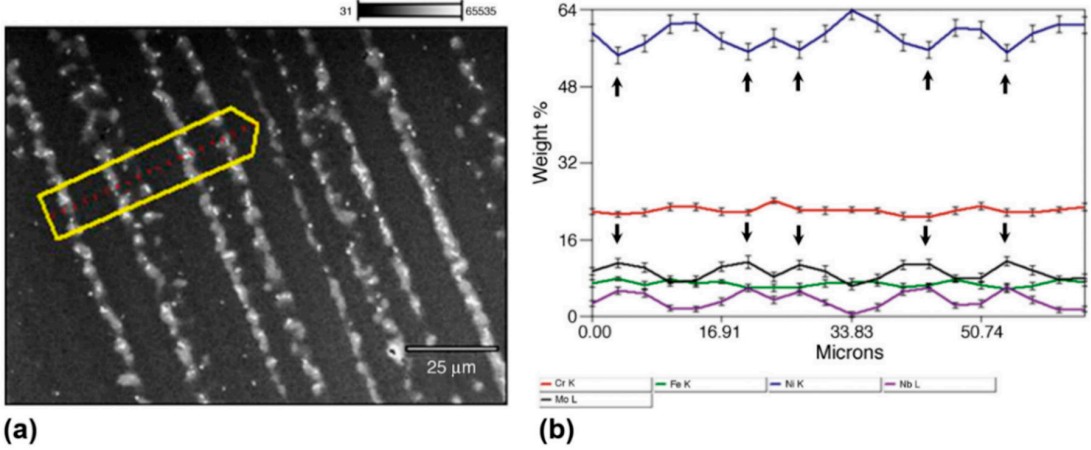

**Figure 9.** (**a**) Elemental chemical profile crossing dendrites, as shown by SEM, and (**b**) elemental segregation during solidification process, as shown by EDS (energy-dispersive X-ray spectroscopy). The arrows indicate the interdendritic regions [51].

Carbon diffusion to the interface area has already been shown to take place during production of the clad pipe. Transportation of the alloying elements may change the phase transformation behavior at the interface. A temperature for the martensite start transformation ($M_s$) is strongly influenced by the chemical composition of the alloy, according to the following, i.e., the Andrews equation [53]:

$$M_s \ (^{\circ}C) = 539 - 423 \, C - 30.4 \, Mn - 17.7 \, Ni - 12.1 \, Cr - 7.5 \, Mo \tag{13}$$

Different $M_s$ temperatures can be used for Fe–Mn–Ni–Cr austenitic weld metal, which can be found in [26].

With a carbon content increase from 0.03 to 0.20 wt % C, the transformation temperature is reduced by around 85 °C. With segregation of the alloying elements, being Ni and Cr in the present case, the $M_s$ temperature will decrease by 37 °C per wt % Ni and 11 °C per wt % Cr. Note that for clad steels, this will be the segregated concentration, not that of the initial bulk.

In addition to the influence from martensite transformation, the enhanced carbon diffusion to the clad side causes extensive precipitation of chromium carbides at austenite grain boundaries in the clad [54]. The precipitation kinetics of $Cr_{23}C_6$ is shown in Figure 10, clearly showing how carbon raises the precipitation temperature, as indicated by the dotted red arrow. With a carbon content of 0.15–0.20 wt % in the clad close to the interface, carbides will spontaneously precipitate as soon as the temperature in hot rolling or diffusion bonding of the clad pipe falls to around 900–950 °C. These reactions may have a negative impact of the corrosion resistance of the clad layer, since part of the Cr content will be consumed by the precipitates.

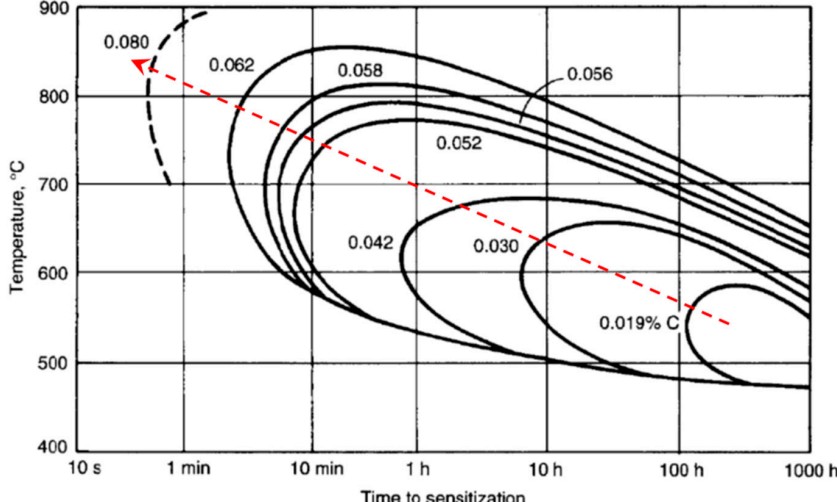

**Figure 10.** Precipitation kinetics of $Cr_{23}C_6$ carbide as a function of carbon content [55]. The dotted red arrow indicates more rapid kinetics due to carbon diffusion from the carbon steel to the clad.

Selection of CRAs for welding wires, such as Alloy 59 or Inconel 625, or stainless steel, will form welds without solid-state phase transformations. This means that the solidification behavior controls the size and shape of grains, microstructure, macro- and microsegregation, distribution of inclusions, defects such as porosity and hot cracks, and ultimately the mechanical properties of the WM, as well as strongly influencing the corrosion resistance.

### 3.3. Weld Metal Microstructures

The WM microstructure in clad pipes is complex as it is a mixture of two different materials (mixed zone) for the root pass (see Figure 11a), the filler wire, and the clad material. For the hot pass (see Figure 11b), the situation can be even more complicated as a third material, the carbon steel, is also incorporated. The use of the term "hot pass" is due to the weld being deposited after the root pass within limited timing. The Schaeffler diagram [56], DeLong [57] diagram, and WRC-1992 [58] diagram have limited use in the prediction of the final microstructure of the WM for Ni-based superalloys due to very high $Ni_{eq}$ values (>55–60 wt %) and low dilution with the carbon steel. The final microstructure will be the Ni-rich $\gamma$-phase. In addition to the chemical composition of the WM, the cooling rate must be considered. It should be noted that the CMT process is often used for root pass welding [59] as a low-heat-input welding process. It tends to provide high cooling rates in the WM, i.e., with $\Delta t_{8/5}$ values even below 2.5 s at atmospheric pressure, which tends to decrease further (to 0.6 s) with increasing pressure [60]. Such fast cooling rates may promote martensite formation.

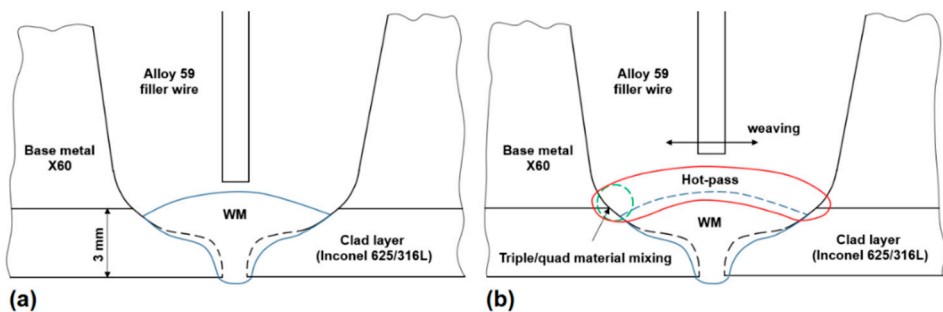

**Figure 11.** (**a**) Root pass with double-material mixing and (**b**) hot pass with triple-material mixing in welding.

### 3.3.1. Root Pass Welding with Ni-Based Wire and Stainless Steel Clad

Electron beam welding (EBW) of Inconel 617 and 310 SS provided a complex microstructure due to nonequilibrium conditions and fast solidification [61]. Due to higher *G/R* values, the WM consisted of small columnar dendrites, which were located near the edge towards the Ni-alloy solid surface. The core of the dendrites consisted of Ni, Mo, and Nb. No cracks or other defects were found. At the weld centerline, a uniform equiaxed cellular structure was obtained. The XRD analyses showed the same γ-Ni matrix lattice in both the base metal and the WM of Inconel 625. The WM hardness was at the same level as Inconel 625 (230 HV), but dropped sharply to 150 HV close to the 304L SS.

Inconel 617 and 310 SS as base metals were joined by applying different filler materials, such as Inconel 82, Inconel 617, and 310 SS. Each filler alloy provided different dilution rates by GTAW [62], and hence different microstructures were formed in the WM. With the use of Inconel 82 wire, the WM was fully austenitic with a substantial amount of dispersed intergranular precipitates (NbC and γ-Laves eutectic). The 304 SS filler wire also provided fully austenitic columnar dendrites. Due to the high $Ni_{eq}$ from the nickel-based superalloy, no δ-ferrite was obtained. The Inconel 617 wire provided fine columnar dendrites, where Mo-rich particles were found in interdendritic regions.

With deposition of Ni-based tubular filler wire on the 316L SS base metal for underwater welding purposes [63], no cracking was observed due to austenite microstructure (5–7% of ferrite is needed to prevent cracking) with equiaxed dendrites with appropriate toughness. However, the tensile test showed a decrease in mechanical properties in WM, corresponding to 85% tensile strength of BM (base metal) with high elongation (up to 32%) and ductile fracture mode, possibly due to lower hardness in the WM. A sharp increase of the hardness was seen near the FL, specifically in the unmixed zone, due to significant iron and some chromium diffusion in from the BM.

Alloy 59 has been used offshore for hot-tapping, while both Alloy 59 and Inconel 625 are candidates for subsea tie-ins and weld repair [13]. These two fillers will give similar weld metal microstructures. The main difference is that the Laves phase and NbC are found in Inconel 625 welds due to the high Nb and C contents, as indicated in Table 1.

### 3.3.2. Root Pass Welding with Stainless Steel Wire and Stainless Steel Clad

The use of AISI 309L SS wire frequently used for the root and hot passes showed a solidification microstructure containing δ-ferrite, a darker phase as illustrated in Figure 12, with a vermicular morphology in the root and columnar morphology in the hot pass [19]. The matrix was austenitic, and this correlates well with the fact that 309L SS is a Cr-rich alloy, in which δ-ferrite is the first phase to solidify. The formation of δ-ferrite (a few percent, e.g., 3–20%) and FA (ferrite/austenite) solidification mode is strongly recommended to avoid solidification cracking in welding with austenitic fillers [64]. A reduction of impurities is very effective in avoiding cracking problems [57].

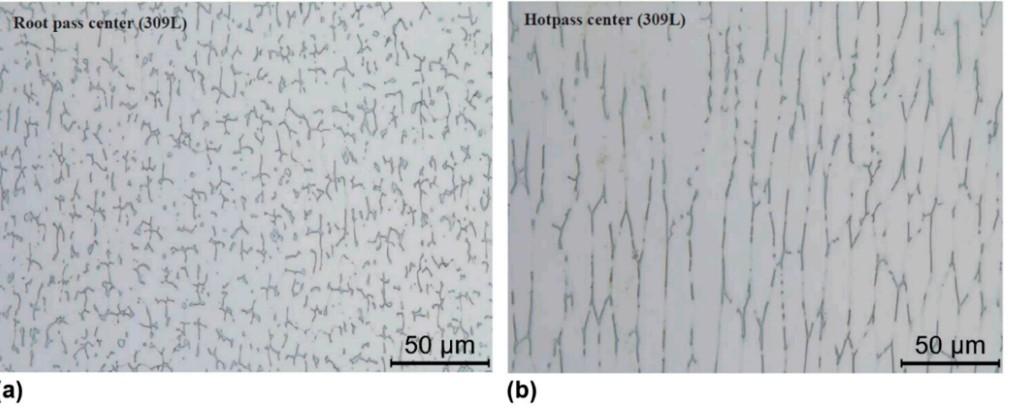

**Figure 12.** Microstructure in the weld metal (**a**) root pass and (**b**) hot pass centers etched with oxalic acid [21].

### 3.3.3. Root Pass Welding with Ni-Based Filler Wire and Inconel 625 Clad

For the combination of a Ni-based filler wire (Alloy 59/Inconel 625) and Inconel 625 clad layer, the weld metal microstructure should not be much influenced by the mixing of clad melting and filler wire due to their similar chemical compositions.

The solidification mode is influenced to some degree by the cooling rate [34]. There is also local variation in chemical composition at the subgrain level due to the microsegregation of alloying (Fe, Cr, Nb, Mo, and Si) and impurity elements (S, P, O, and C) during solidification, leading to the second phase formation (e.g., carbides and Laves phase for Inconel 625). These phases form at the end of solidification in the interdendritic regions. Composition analyses show that dendrite cores consist of Ni, whereas interdendritic areas mostly include the precipitation of Nb (carbides, Laves phase) and Mo (Laves phase) [65]. An example is shown in Figure 13. The Laves phase is an intermetallic compound of the type $A_2B$, where A is Ni or Cr and B is constituted by Mo or Nb.

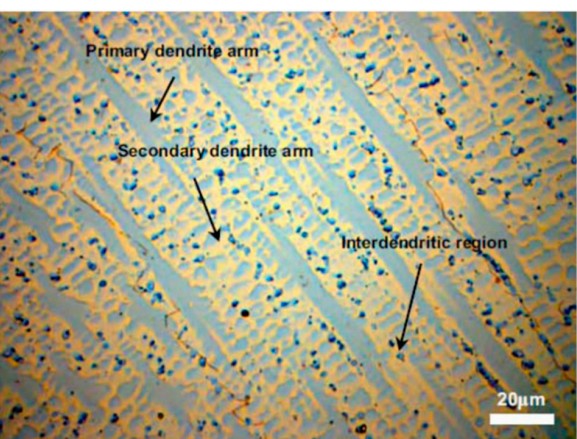

**Figure 13.** Primary weld metal microstructure with Inconel 625 filler wire.

Due to the segregation of alloying and impurity elements, there is a possibility for liquation or solidification cracking (Figure 14a) to occur in the welding of solid-solution-strengthened alloys, as well as ductility-dip cracking (DDC); see Figure 14b. DDC is particularly common for multipass welding due to reheating [65]. Regarding nickel-based alloys, Ren et al. [66] identified that the constitutional liquation of $M_{23}(B,C)_6$ (M can be Mo, Cr, or Fe) carbides is responsible for the liquation cracking in WM. It was noted that the liquation cracking was reduced with a decrease in heat input; the heat input must, therefore, be optimized during welding.

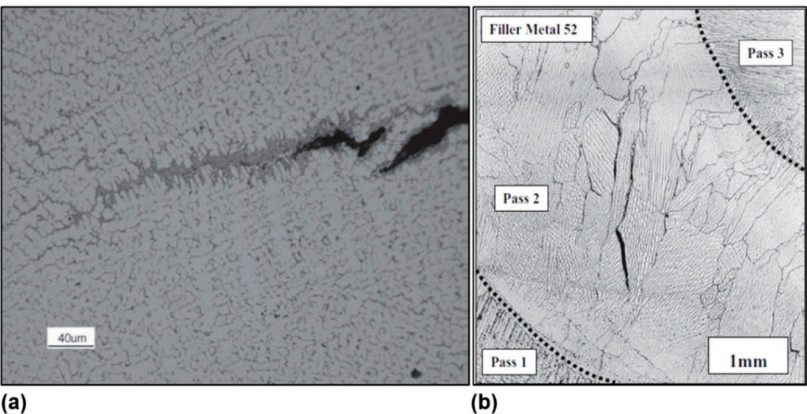

**Figure 14.** (**a**) Solidification crack: the crack path and eutectic constituents are coexisting. The eutectic constituent forms from the terminal solute-rich liquid at the end of solidification. (**b**) Ductility-dip cracking (DDC) in multipass welding [65].

Recently, it was found that significant reduction (by ~90 vol %) in Laves phases in welds can be achieved by applying an acoustic device which induces the vibration of the weld pool through a high-frequency current (the ultrasonic Ampere's force) at a specific frequency rate [67]. Here, two mechanisms are involved during ultrasonic treatment: reduction of the local concentration of Nb and specific (distorted) Ni-dendrite growth directions.

### 3.3.4. Hot-Pass Welding

In the hot pass, the three materials are mixed (as visualized in Figure 11b), namely X60 (BM), Ni-based Alloy 59 filler wire, and clad material (316L SS). Some part of the previously deposited root pass will also be melted, forming a quaternary mixing system. The dilution rate will depend on both welding parameters and bevel geometry, particularly root gap and root face height. With excessive carbon diffusion from X60 or from the carbon-enriched interface between the clad and BM of the pipe, there is a risk of embrittlement.

The chemical composition of single beads was determined by EPMA to include both root and hot pass, as shown in Figure 15. For a 309 SS wire, there is no significant change in the chemical composition. When using Inconel 625 wire (see Figure 16), Ni increases significantly in the root pass, while Fe decreases. This is related to the melting of a part of the pipe steel (X60) which has low Ni content. For the same reason, a slight increase in Cr, Mo, and Nb in the root pass is also observed, while the Fe concentration is substantially lower in the root pass, clearly showing very low dilution with the pipe. On the contrary, the hot pass contains 20–30 wt % Fe, which implies a certain dilution with the pipe.

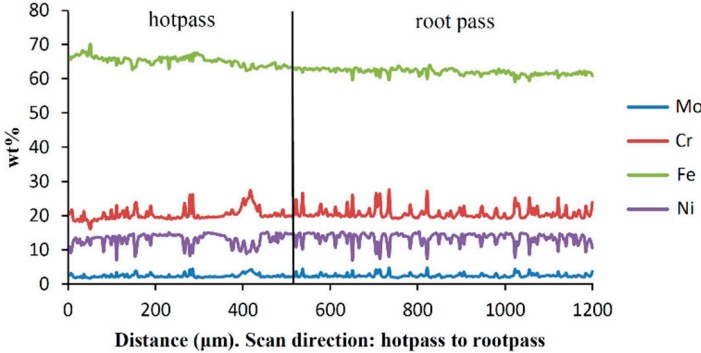

**Figure 15.** Electron probe microanalyzer (EPMA) quantitative analysis of X60 steel pipe with AISI 309 SS filler wire [19].

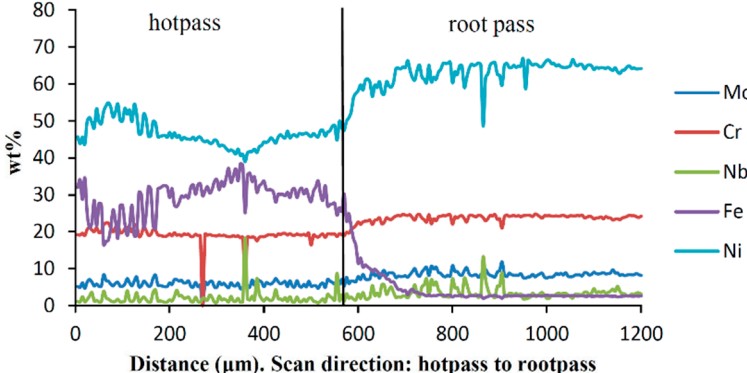

**Figure 16.** EPMA quantitative analysis of X60 steel pipe with Inconel 625 filler wire [19].

### 3.4. Heat-Affected Zone

The HAZ of the pipeline steel is another important aspect to consider, due to the grain growth near the FL. As a result, the coarse-grained HAZ (CGHAZ) should be considered. The major parameters which may control the final microstructure evolution of the HAZ are the weld cooling rate, the clad pipe production method (e.g., TMCP or QT steel) [68], and the chemical composition of the material [65].

### 3.4.1. High-Strength, Low-Alloy Steel HAZ

Modern pipeline steel normally has low carbon content (≤0.06 wt % C). The final HAZ microstructure can be predicted by a CCT (continuous cooling transformation) diagram, e.g., for X70 steel (see Figure 17), where it can be seen that a fully martensitic microstructure is formed upon fast cooling ($\Delta t_{8/5} < 1.4$ s) with a hardness level of ~340 HV. At slower cooling rates (>20 s), coarse bainite (~220 HV) is formed. This type of microstructure may give low toughness at subzero temperatures.

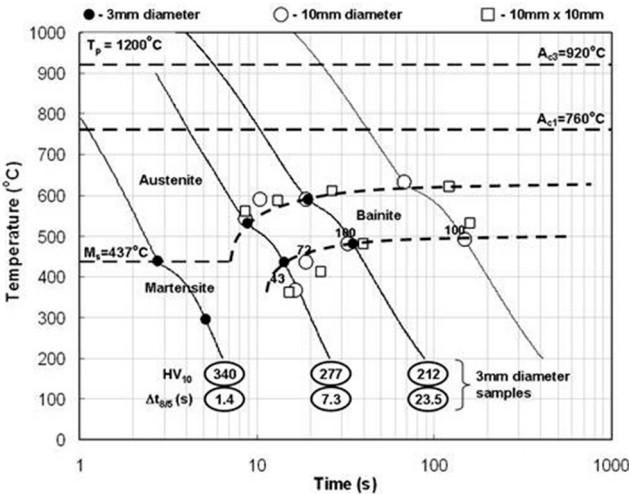

**Figure 17.** CCT (continuous cooling transformation) diagram for X70 steel [69].

In the case of multipass welding (filling passes), reheating of the CGHAZ may induce the formation of brittle martensite–austenite (M-A) islands, i.e., the intercritically reheated CGHAZ (ICCGHAZ). Hence, very low toughness at subzero temperatures may occur [70,71]. Afterwards, a comprehensive study by Mohseni et al. [72–75] and Haugen et al. [76] on brittle crack initiation in the ICCGHAZ confirmed earlier results [77].

### 3.4.2. HAZ of Welded 316L Stainless Steel Clad

The HAZ of austenitic stainless steels (e.g., 304L/309/316L) consists of an austenitic matrix after solidification with almost negligible grain growth when compared to carbon steels. The grain size of the 316L SS base metal is ~45–75 µm, where the variation highly depends on manufacturing route [78]. When compared to fine-grained carbon steel having grain size smaller than 10 µm [79], there is a much lower driving force for grain growth [80]. In addition, upon cooling, some delta ferrite may form at the HAZ grain boundaries, suppressing grain growth [57]. This results in a minor hardness increase near the FL [63]. However, despite the low susceptibility for grain growth, high heat input should be avoided, especially when the BM has smaller grains that can cause significant grain growth even in austenitic steels with subsequent softening [81]. In addition, due to high heat input, there is the formation of chromium carbides on the grain boundaries of austenite. As a result, the area around the carbides has lower chromium content compared to the bulk material. This can lead to the sensitization phenomenon, which reduces corrosion resistance significantly [57].

Due to dissolved precipitates, some carbides and nitrides can form upon cooling along grain boundaries in the HAZ or at the ferrite–austenite interface, especially when the stainless steel contains

Ti and Nb. This can cause grain boundary liquation by lowering the melting temperature, in addition to the segregation of impurities (primarily P and S), causing liquid films upon cooling. Notably, between the FZ and the HAZ, there is the partially melted zone (PMZ) due to temperature gradients, which has low mechanical properties [82]. Grain growth in this zone is not significant and the grain size is similar to the as-received BM [83]. The thickness of the UMZ (unmixed zone) (<0.5 mm) can be reduced by applying mechanical vibrations during welding [84].

### 3.4.3. HAZ of Ni-Based Superalloy Clad

Ni-based superalloys are divided into solid-solution-strengthened (substitutional alloying elements: Cr, Fe, W, Cu, Mo, Co, Ta, and Re; among them are 59, 617, 625, 690, 800, 825, and 902 alloys [65]) and precipitation-strengthened (precipitation formation elements: Ti, Al, and Nb; among them are 300, 263, 713, 718, 725, 751, 925, 945, and Rene 41 alloys [65]) categories, the latter of which are not frequently used for cladding. Therefore, further metallurgical aspects for solid-solution-strengthened alloys only are discussed.

Due to the high alloy level of Inconel 625 (see Table 1), additional metallurgical reactions may take place when it is used as a clad. In the HAZ, these may consist of the dissolution and reprecipitation of NbC, grain boundary segregation, and liquation phenomena [65]. A segregation mechanism of some elements (S, P, Pb, and B) has the most profound effect on liquation in the HAZ, and the principal mechanisms are illustrated in Figure 18. The same mechanisms are valid for stainless steels.

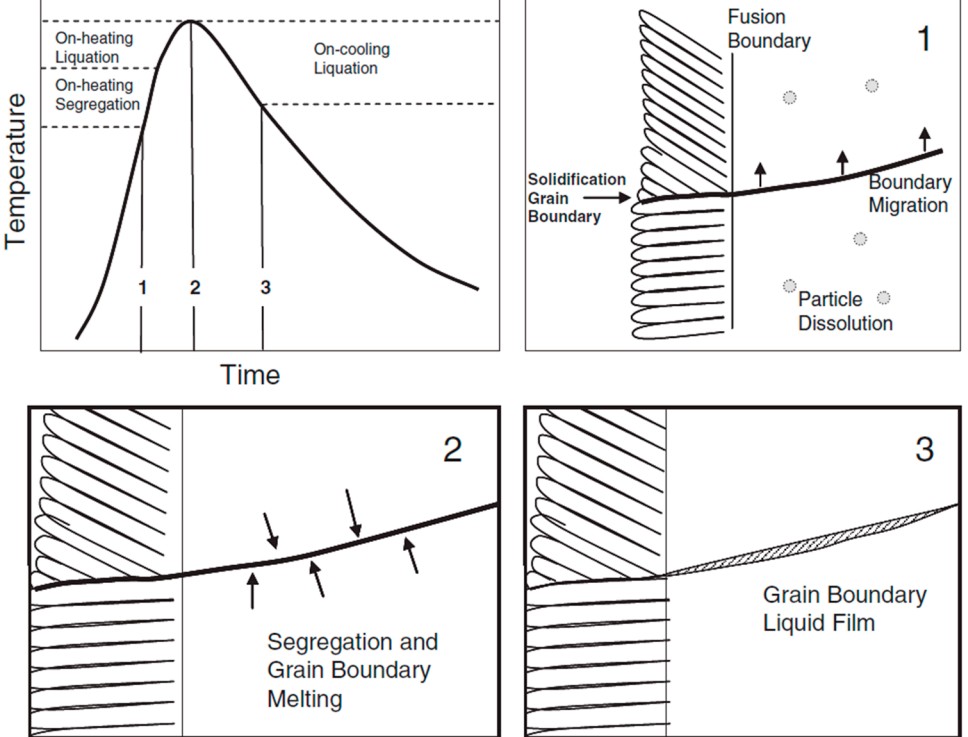

**Figure 18.** Solute segregation and localized melting along grain boundaries in HAZ during different temperature phases in welding [65]; 1, 2, and 3 define progression phases of liquation cracking.

Grain growth in welding of these alloys does not present any problems, since the initial grain size is coarse (~20–40 μm [85]). Accordingly, the driving force for grain growth is very low and similar to stainless steels. A secondary factor is the interaction of secondary phases (e.g., borides, sulfides, and carbides such as TiC and NbC) with grain boundaries by a constitutional liquation [86]; this is called penetration mechanism [87]. During welding (rapid heating), secondary phases do not dissolve within the matrix, and upon heating, they react with the matrix by forming interfacial

liquid films [65]. However, small-scale $M_{23}C_6$ particles dissolve at high temperatures during GTAW welding [88]. NbC and TiC can also be responsible for hardness and strength increases, based on experimental observations made by GTAW of 2 mm Inconel 625 [89]. As a result, a high-heat-input process must be avoided.

The liquation cracking may occur in the partially melted zone (PMZ) in the HAZ adjacent to the FL. The crack length has been observed to increase with a decrease in heat input [66]. Preheating suppressed the crack length, possibly by reducing the cooling rate. Therefore, the cooling rate must be adjusted to avoid the liquation cracking.

Another factor contributing to crack forming is stresses built up during solidification and further cooling [65].

### 3.4.4. Bimetallic Interface After Welding

The interface between the 316L SS clad and high-strength, low-alloy (X60/X65) base metal has already been discussed in Section 2, as the majority of the elemental diffusion and segregation has taken place during the clad pipe production. In hot-roll (metallurgical) bonding of the clad pipe, carbon diffuses at elevated temperatures ($>A_3$ point in the Fe–C diagram) from the carbon-rich ferrite to the austenite, with subsequent carbide precipitation of $M_{23}C_6$, mainly $Cr_{23}C_6$ [90]. This diffusion results in a softer, carbon-depleted zone in the ferrite (decarburized layer in the pipe steel close to the interface), and a hard, carbon-enriched zone (carburized layer) in the 316 SS (austenite) clad [91]. Moreover, this phenomenon makes the clad sensitized, and the existence of Cr-depleted zones may promote intergranular corrosion [92].

After welding, interface cracking has been found between the BM (X60) and the 316L SS clad material, as shown in Figure 19a. It was observed that the cracking was usually initiated in the HAZ. Upon further inspection, it was found that there is a grain boundary precipitation on the clad side near the interface (see Figure 19b) [93]. The electron microprobe analysis revealed high carbon content in the clad near the interface, which increases the risk of cracking [21]. This possibly takes place due to additional stresses caused by welding; i.e., residual stresses added to production thermal stresses because of thermal expansion mismatch between the 316L stainless steel clad and the pipe steel. Fortunately, the cracks occurred in the direction following the pipe length, where the load is low, and were not regarded as representing any risk of pipeline failure. The cracking problem can be prevented using a Ni interlayer [17]. To the authors' knowledge, similar cracking problems with Inconel 625 have not been reported.

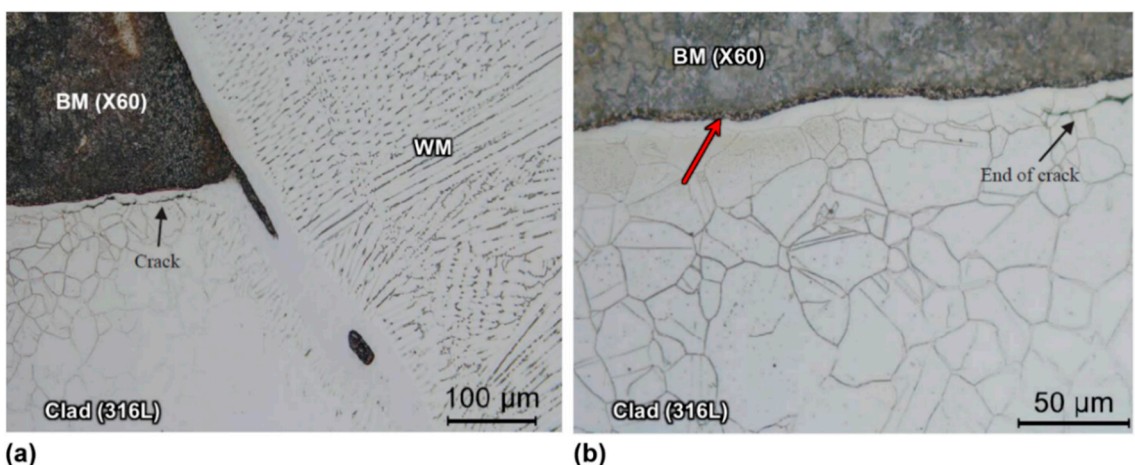

**Figure 19.** (**a**) Cracking near the bimetallic interface in the clad and (**b**) grain boundary precipitates, as indicated by the red arrow, near the interface [21].

## 4. Potential Application of Hyperbaric Weld Repair

Repair welding of clad and lined pipes requires advanced and robust hyperbaric welding processes. It is thus critical to understand the physical phenomena of the welding process as well as optimize the welding parameters to obtain high-quality welds. During the last decade, significant progress has been made in the field of modeling of the heat and mass transfer during GMAW. Models including the computation of current density, temperature, and momentum transfer, including the transfer of individual droplets, have been presented [94–105].

The complex interaction of the shielding gas and current with weld bead appearance and temperature distribution has been studied. However, no detailed modeling of the arc physics and metal transfer under hyperbaric conditions has been found in the literature. One could expect that both the arc pressure and the effect of droplet impingement would be stronger in the case of a more *confined* or *constricted* arc with higher shielding gas density. There are five main flow-driving forces during arc welding processes that should be investigated: the gradient of surface tension (Marangoni force), the gas drag force at the surface, the buoyancy, the electromagnetic (Lorentz) force, and the moment transferred by impinging droplets in the weld pool. In the case of CMT, the droplet momentum is strongly reduced, but there could be other challenging effects from the large variations in current and changes in the boundary conditions for the plasma arc. So far, no papers with numerical simulation of the CMT arc mode physics have been published, which is probably related to the high complexity of the process.

## 5. Conclusions

The present work has addressed metallurgical changes in the welding of clad pipes, and several conclusions may be drawn:

- Complex metallurgical phenomena present in the welding of clad pipes due to multi-material systems, especially concerning the weld metal. Clad pipes represent a solid economic alternative to pipes made of stainless steel which is vital to the oil and gas industry.
- Welding process significantly alters the base metal near the weld metal, i.e., the heat-affected zone. Therefore, in complex clad pipes, utilizing expensive materials must be considered in detail.
- Solidification parameters have very high importance on solidification behavior and resulting mechanical properties. Moreover, they can be efficiently estimated by numerical simulation at any point in welded joints.
- The subsea clad pipe network requires an emergency method for pipe repair to be developed. Therefore, the current manuscript represents an important contribution to the knowledge on microstructural changes and diffusion that may occur in welding, and how these may influence the pipeline integrity.
- During repair under hyperbaric conditions, the use of multiple filler wires is very complicated for clad pipes. Therefore, the suitability of filler wire for multi-material systems, both for carbon and nickel-based/stainless steel materials, must be carefully investigated for single filler wire applications.

**Author Contributions:** I.B. wrote the manuscript and performed numerical simulations. V.O. and O.M.A. supervised the further improvement of the manuscript.

**Funding:** The authors wish to thank the Research Council of Norway for funding through the Petromaks 2 Programme, Contract No. 234110/E30. The financial support from Statoil Petroleum AS, Gassco AS, Technip Norge AS, EDF Induction AS, and Pohang Iron and Steel Company Posco is also acknowledged.

**Conflicts of Interest:** The authors declare no conflict of interest.

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
