# Peer review of "Metallurgical Aspects in the Welding of Clad Pipelines—A Global Outlook"

_applsci, doi:10.3390/app9153118_

Round 1

Reviewer 1 Report

The Paper is very well conceived. Congratulate!!

Author Response

Reviewer had no questions and expressed very positively about the manuscript.

Reviewer 2 Report

This article, without a description of the experimental process, is not a typical research paper and is more like a review paper.
There are a few suggestions listed below:
Line 181 Table 1:
It is recommended to provide the chemical compositions of all materials described in this article.
Line 106-107
The method and apparatus for measuring carbon content should be provided.
Line 107-108
The location of the grain growth in this study should be indicated.
Line 160-161
The Cr and Ni equivalents only can be calculated with Equation 2.
Line 256
Figure 6(a) is different from the schematics diagram of weld design in Figure 11. The base metal seems to be carbon steel. Figure 6(a) is not in line with Figure 6(b).
Line 260-262 "At the WM centerline, the temperature gradient is smaller and the solidification rate faster when compared to the area near the FL (a liquid/solid interface)."
This sentence seems to be not idetical to the results of Tables 3 and 4.
Line 443-449
Solidification cracks and DDC are classified into two different crack mechanisms. DDC belongs to a reheat cracking. A nickel-based alloy containing fewer impurities elements such as Nb and Ti is more susceptible to DDC.

Author Response

Point 1: This article, without a description of the experimental process, is not a typical research paper and is more like a review paper.

Response 1: The paper, indeed, contains extensive literature review due to complexity of the selected topic, with original work presented.

Point 2: Line 181 Table 1: It is recommended to provide the chemical compositions of all materials described in this article

Response 2: The table was expanded including typical chemical composition of high strength low alloy steel, and 316L stainless steel

Point 3: Line 106-107: The method and apparatus for measuring carbon content should be provided

Response 3: Carbon content was measured by EPMA. Relevant note was added to the manuscript.

Point 4: Line 107-108: The location of the grain growth in this study should be indicated.

Response 4: Grain growth was near interface as written in manuscript.

Point 5: Line 160-161: The Cr and Ni equivalents only can be calculated with Equation 2.

Response 5: The numbers of equation were fixed.  

Point 6: Line 256: Figure 6(a) is different from the schematics diagram of weld design in Figure 11. The base metal seems to be carbon steel. Figure 6(a) is not in line with Figure 6(b).

Response 6: Figure 11 represent only schematical diagram regardless the shape of weld metal. The maximum possible geometry was achieved by numerical simulation and simulated line is fairly in good agreement with experimental results. More than 50 numerical simulation were performed.

Point 7: Line 260-262: "At the WM centerline, the temperature gradient is smaller and the solidification rate faster when compared to the area near the FL (a liquid/solid interface)." This sentence seems to be not identical to the results of Tables 3 and 4.

Response 7: The sentence is correct and true for the given tables. It was modified to avoid misunderstanding: for each selected material.

Point 8: Line 443-449: Solidification cracks and DDC are classified into two different crack mechanisms. DDC belongs to a reheat cracking. A nickel-based alloy containing fewer impurities elements such as Nb and Ti is more susceptible to DDC.

Response 8: Relevant text was slightly updated. In this case the text is relevant only for nickel-based alloys and no comparison with stainless steels. Nb, Ti are more alloying elements not impurity elements.

Reviewer 3 Report

Dear Authors,

I am afraid the article is not a research article but the review one. However, it is labelled as 'Article'. In that case I do not like its structure. In the form it is now there is almost impossible to estimate which parts are the literaturę overview and which are the original your input. The numer of references is big and the citations are present in all chapters. Please re-organize the article because in that form I recommend to not publish it.

Sincerely,

Reviewer 

Author Response

Remaking structure of the article is not possible at this stage and we strongly believe it is optimizied due to many revisions we made previously. The article, indeed, contains very extensive literature review since the presented topic is complex. The structure was slightly re-organized based on other reviwers' points.

Reviewer 4 Report

This a real state of the art about techniques to weld pipelines, written by one of the most important research centers in the world. At OMAE conferences they show the more cutting edge advances in welding, and this paper is a good summery.

Several aspects can be improved…a little:

Conclusions: establish some ideas about the trends in the short and in the long run scenarios.

Please, provide some more information about the welding process itself to complete the documentation of the whole process.

Some facts and figures related to the industry in which this kind of pipes were used should be included in the introduction in order to present a clear industrial context, and scientific relevance.

Author Response

Point 1: Conclusions: establish some ideas about the trends in the short and in the long run scenarios.

Response 1: Conclusions were updated.

Point 2: Please, provide some more information about the welding process itself to complete the documentation of the whole process.

Response 2: More extensive explanation of welding process was added in introduction. 

Point 3: Some facts and figures related to the industry in which this kind of pipes were used should be included in the introduction in order to present a clear industrial context, and scientific relevance.

Response 3: We strongly believe introduction contains important aspects of clad pipes welding and it is not necessarily to add more figures to the manuscript. Introduction was also expanded based on other reviewers' comment.

Round 2

Reviewer 2 Report

Accept in present form

Reviewer 3 Report

I am sorry but I do not accept your answer. So, I confirm my previous decision and recommendation.

Sincerely,

Reviewer